# Squamotransitional Cell Carcinoma of the Uterine Cervix with Ovarian Metastasis and Benign Brenner Tumor: A Case Report

**DOI:** 10.3390/reports6040054

**Published:** 2023-11-13

**Authors:** Angel Yordanov, Milen Karaivanov, Ivan Ivanov, Stoyan Kostov, Venelina Todorova, Ilko Iliev, Eva Tzoneva, Diana Strateva

**Affiliations:** 1Department of Gynaecological Oncology, Medical University Pleven, 5800 Pleven, Bulgariastrateva_d@abv.bg (D.S.); 2Department of General and Clinical Pathology, University Hospital “Dr. Georgi Stranski”, 5800 Pleven, Bulgaria; mkaraivanov1962@gmail.com (M.K.); posledenzalez@gmail.com (I.I.); 3Department of Gynecology, St. Anna University Hospital, Medical University—Varna “Prof. Dr. Paraskev Stoyanov”, 9000 Varna, Bulgaria; drstoqn.kostov@gmail.com; 4Imaging Department, University Hospital “Dr. Georgi Stranski”, 5800 Pleven, Bulgaria; venelina.todorova93@gmail.com; 5“Dr. Shterev” Hospital, 1000 Sofia, Bulgaria; dretsoneva@gmail.com

**Keywords:** squamous cell carcinoma, papillary squamotransitional cell carcinoma, ovarian metastasis, Brenner tumor, endometrial invasion

## Abstract

Introduction: Cervical cancer is the fourth most common malignancy in women and the fourth leading cause of death among women. The main histological types of cervical cancer are squamous cell carcinoma—75% of all cases; adenocarcinoma—10–25%; and all other rare variants including adenosquamous carcinoma and neuroendocrine carcinoma. Squamotransitional cervical cancer is an extremely rare and poorly studied subtype of squamous cell carcinoma. Case report: We present a case of a 64-year-old female patient with early-stage squamotransitional carcinoma. A metastasis was observed in the left ovary and the left fallopian tube and a benign Brenner tumor in the right ovary. Discussion: Although it is believed that this cervical cancer subtype shares the same risk factors and prognosis as squamous cell carcinoma, it is more likely to metastasize and recur. It is not unusual for spread to exist within nearby structures like the cervix and adnexa. It is impossible to tell which is the predominant focus from the immunoprofile of the lesions. Practically speaking, the best course of action in these situations is to rule out the presence of a primary tumor in the urinary tract before clarifying the condition of the cervix, uterus, and adnexal tissues. The presence of a Brenner tumor raises the possibility of a connection between the tumor’s differentiation from a cell population and potential urothelial differentiation. Conclusion: Squamotransitional cervical cancer is a rare tumor with a poorly studied clinical behavior. Despite a shortage of information in the literature, it should be regarded as a more aggressive variety of squamous cell carcinoma and, as such, should be treated and followed up more aggressively. This case is the first described with involvement of the cervix, endometrium, and adnexal structures and a concomitant Brenner tumor.

## 1. Introduction

Due to the deployment of vaccinations and screening programs, the incidence of cervical cancer (CC) is declining globally, but mainly in developed countries [1]. But it is also the fourth most common cause of mortality for women and the fourth most prevalent malignancy [2].

Squamous cell carcinoma (SCC), which accounts for 75% of all cases, adenocarcinoma (10–25%), and all other uncommon variants, such as adenosquamous carcinoma and neuroendocrine carcinoma, are the main histological forms of CC [3]. The subtypes of SCC are classified as keratinizing, non-keratinizing, basaloid, verrucous, warty, papillary, lymphoepithelioma-like, and squamotransitional, each with a variable frequency [4]. Squamotransitional CC is incredibly uncommon and poorly understood. It is assumed that its frequency is about 1.6% of all cervical malignancies [5]. This tumor is thought to have an aggressive course and can develop early metastases to the ovaries [6].

We present a case of a 64-year-old female patient with early-stage squamotransitional carcinoma in which metastasis was observed in the left ovary and left fallopian tube and a benign Brenner tumor in the right ovary.

## 2. Detailed Case Description

A 64-year-old female patient with histologically verified CC because of menopausal bleeding was admitted to the clinic. The patient has been amenorrhoeic for 20 years, reports arterial hypertension, glaucoma, appendectomy, and two normal births. The cervix was clearly intact, and there were no abnormalities found during the gynecological examination. On the ultrasound exam, a 2/2 cm formation with unclear borders in the cervical canal was found. Chest X-ray and blood tests came out ordinary.

A cervical tumor measuring 1.89/1.78 cm was seen on an MRI without lymph node involvement or parametrial invasion. In the right ovary, a 1.03/1.12 cm mass was visible but uninterpretable (Figure 1).

The patient was staged as T1B1N0M0 and a type C2 radical hysterectomy according to Querleu–Morrow Classification with total pelvic lymph node dissection was performed [7].

The morphologic finding showed a cervix infiltrated by a nested tumor composed of cells with vesicular nuclei, condensed chromatin, and moderate eosinophilic cytoplasm, with propagation in the cavum uteri, and the immunoprofile of the lesion favored squamous urothelial carcinoma (Figure 2). The presence of tumor emboli was observed. The proximal 1/3 of the vagina and the vaginal resection line were free of tumor infiltration.

In the left ovary, the following were observed: part of the fallopian tube with corpus albicans, serous simple cysts, and infiltration by a nest-infiltrative tumor composed of cells with vesicular nuclei with condensed chromatin and pale eosinophilic cytoplasm. Numerous mitoses were observed in the tumor population (Figure 3).

Left fallopian tube with atrophic changes and tumor infiltration (Figure 4).

In the right ovary, a small benign Brenner tumor was found on a background of corpus albicans (with an immunoprofile showing positivity for GATA3, CK7, and CK 5/6, and negative for CK20 and WT-1) (Figure 5). The right fallopian tube showed atrophic changes.

In all 16 removed lymph nodes, no tumor metastases were found.

The patient was staged as pT1B1pN0M0 according to the 8th TNM classification and 1B1 FIGO stage according to the 2021 update.

The patient was discharged on postoperative day 7 without complications. She was directed to conduct chemoradiation, having only performed percutaneous radiotherapy—56 Gy. Five months after the surgical treatment, there was no evidence of a recurrence of the disease.

## 3. Discussion

Transitional cell carcinomas were originally documented in the ovaries and fallopian tubes, and then in the endometrium and cervix [8,9,10], with Marsh providing the earliest description of these cancers in the cervix in 1952 [11]. The World Health Organization (WHO) cervical cancer histological classification recognized this form of carcinoma as a distinct subtype in 2003 [12], following Randall et al.’s [5] proposal from 1986.

Koenig et al. divided squamotransitional CC into three types depending on the papillary component, each with a different frequency [13]:Predominantly squamous (28.1%);Mixed squamous and transitional (50%);Predominantly transitional (21.9%).

Although it is believed that this subtype shares the same risk factors and prognosis as SCC, it is more likely to metastasize and recur [14,15]. Even a few months after finishing treatment, there have been examples of early recurrence reported in the literature, which is indicative of a more aggressive course [15,16].

Such tumors with different names (“transitional cell carcinomas”/“squamous-urothelial carcinoma”/“Papillary squamotransitional carcinoma”), sharing somewhat similar features, are described in the literature. They may involve a single site or involve more than one structure of the female reproductive system simultaneously, demonstrating somewhat consistent immunoprofiles (Table 1).

Our data support the statement by Patrelli TS et al. [18] that leading factors for the diagnosis are the morphology and cytoarchitectural features of this type of tumor. The morphological findings are consistent with mitotically active tumors with solid and/or inverted papillary growth patterns, composed of polygonal cells demonstrating transitional cell features (with hyperchromatic, oval, grooved nuclei and a relatively small amount of cytoplasm) showing focal squamous differentiation. Additionally, immunohistochemical staining with CK7 (usually positive) and CK20 (always negative) are good auxiliary markers for the diagnostic process. The authors also suggest other immunomarkers including p63 and p16ink4a for the differential diagnosis.

Analyses based on larger populations demonstrate the almost persistent CK7-positive, CK20-negative (with rare exceptions) profile of squamotransitional cell carcinomas of the uterine cervix [13].

According to a large study of CK7 and CK20 expression among different tumors, Brenner tumors demonstrate a CK7-positive, CK20-negative immunoprofile [21].

At the same time, Brenner tumors are recognized to be diffusely immunopositive for GATA3 [22].

It is common knowledge in clinical practice that transitional cell carcinoma of the endometrium can coexist with ovarian Brenner tumors, raising the possibility that there may be some relationship between the two. It is hypothesized to be either a result of “multicentric metaplastic process” (neometaplasia), affecting the ovarian coelomic epithelium and Mullerian uterine epithelium, or consequence of the so called “field effect”—site-determined presentation of an entity [23]. We consider both hypotheses as possible in the context of the presented case.

Because urothelial tumors are a highly variable group, attempting to compare the immunoprofile of transitional cell carcinomas with those of urinary tract origin is not a wise choice [24].

GATA 3 immunoexpression is not a reliable marker to delineate urothelial differentiation in the context of tumors of the female reproductive system, especially in view of the fact that it is expressed in half of squamous cell carcinomas of the cervix [25].

Drew PA et al. concluded that squamous-urothelial carcinomas of the cervix are more closely related to squamous carcinomas based on an immunoexpression investigation for p63, p16INK4a, and uroplakin 3 [26]

A subset of uncommon tumors that provide a significant problem in differential diagnosis in clinical practice are cervix-related tumors that affect the urinary system secondarily and are hard to identify from urothelial carcinomas. It is exceedingly challenging to distinguish them from urothelial carcinomas since they share morphological traits with squamous and urothelial carcinoma. Ten comparable malignancies were found by Schwartz LE et al. based on a review of consultation case archives from Johns Hopkins Hospital for the years 1984 to 2016. In their immunohistochemistry test for p16 and GATA3, 8/10 of the samples were positive for p16, whereas only 6/10 of the samples tested negative for GATA3. In situ hybridization was used to test for the presence of HPV, and in 8 out of 10 instances, high-risk HPV was found. The authors emphasize the importance of clinical manifestations and organ findings [27].

In conclusion, tumors that resemble transitional cell carcinoma in terms of form and immunoprofile frequently affect the cervix, endometrium, and ovaries. It is not unusual for spread to exist within nearby structures like the cervix and adnexa. It is impossible to tell which is the predominant focus from the immunoprofile of the lesions. Practically speaking, the best course of action in these situations is to rule out the presence of a primary tumor in the urinary tract before clarifying the condition of the cervix, uterus, and adnexal tissues.

The presence of a Brenner tumor suggests a possible relationship with the occurrence of non-differentiation of the tumor from a cell population with possible urothelial differentiation.

The immunoexpression of the tumor we described, the negativity for GATA3, and the morphology of the lesion may suggest that such tumors could possibly be assigned to squamous cell carcinomas, but possess a peculiar morphology and growth characteristics that should be further investigated in the context of potential clinical applications.

Finally, according to the contemporary WHO classification of the tumors of the uterine cervix, squamotransitional carcinoma is a subtype of squamous cell carcinoma associated with HPV. The variant is defined as having “transitional- like appearance “, but no immunoprofile is recommended. It is currently considered a histologic type from the past [28]. Still, the morphologic presentation of the lesion and the differential diagnosis with involvement from transitional carcinoma originating from the urinary tract makes this morphologic pattern important for clinical and diagnostic practice. No definitive immunoprofile is recommended for squamotransitional carcinoma, but the above-mentioned CK7+/CK20− profile is significant for distinguishing it from transitional carcinoma infiltration/metastatic involvement (see Table 1). Follow up of cases of squamous cell carcinomas with morphologic squamotransitional patterns is important to assure that such tumors have similar prognosis to other HPV-related squamous cell carcinomas with no specific morphologic features and to monitor if the optimal therapy is applied.

## 4. Conclusions

Squamotransitional CC is a rare tumor with a poorly studied clinical behavior. Despite the paucity of data in the literature, it should be considered a more aggressive variant of SCC and as such should be more aggressively treated and more actively followed up. This case is the first to describe involvement of the cervix, endometrium, and adnexal structures and a concomitant Brenner tumor.

## Figures and Tables

**Figure 1 reports-06-00054-f001:**
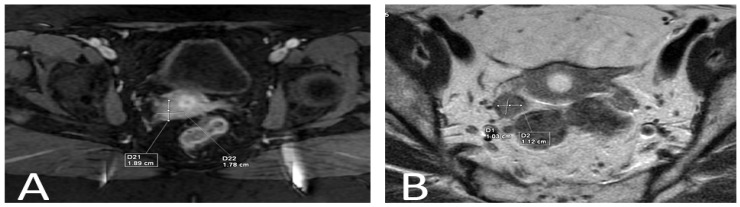
(**A**) Finding in the cervix; (**B**) finding in the right ovary.

**Figure 2 reports-06-00054-f002:**
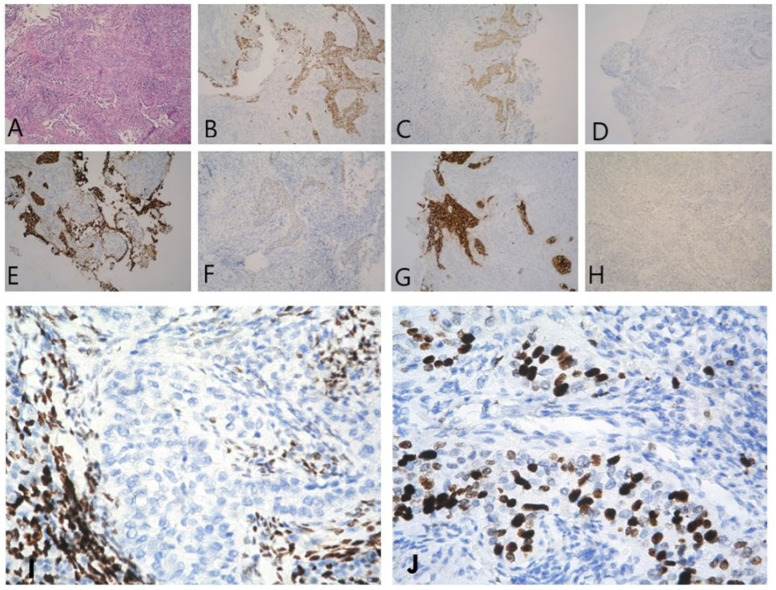
(**A**) Nests and tufts of tumor cells infiltrating and obliterating the endocervical mucosa and endometrium XE, 100×; (**B**) CK5/6—positive in tumor cells, 100×; (**C**) p16—moderate-intensity nuclear positivity and weak cytoplasmic IHC expression in tumor cells, 100×; (**D**) GATA 3—negative, 100×; (**E**) CK7—positive in tumor cells, 100×; (**F**) CK20—faint IHC expression in tumor cells, 100×; (**G**) CEA—positive in tumor cells, 100×; (**H**) p63—weak nuclear IHC expression in about 60% of p63 tumor cells, 100×; (**I**) ER—negative in the tumor cells in the context of positive stromal cells, 400×; (**J**) Ki-67—high proliferation rate in the invasive tumor nests, 400×.

**Figure 3 reports-06-00054-f003:**
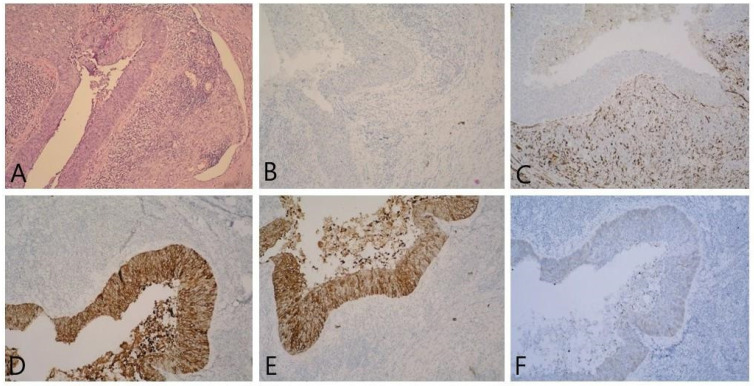
(**A**) Ovarian parenchyma with nests of tumor cells showing cystic transformation HE, 100×; (**B**) GATA 3—negative, 100×; (**C**) WT-1—negative in tumor cells against a background of WT-1-positive stroma, 100×; (**D**) CK7—positive in tumor cells, 100×; (**E**) CK5/6—positive in tumor cells, 100×; (**F**) weak IHC expression in CK20 tumor cells, 10.0×.

**Figure 4 reports-06-00054-f004:**
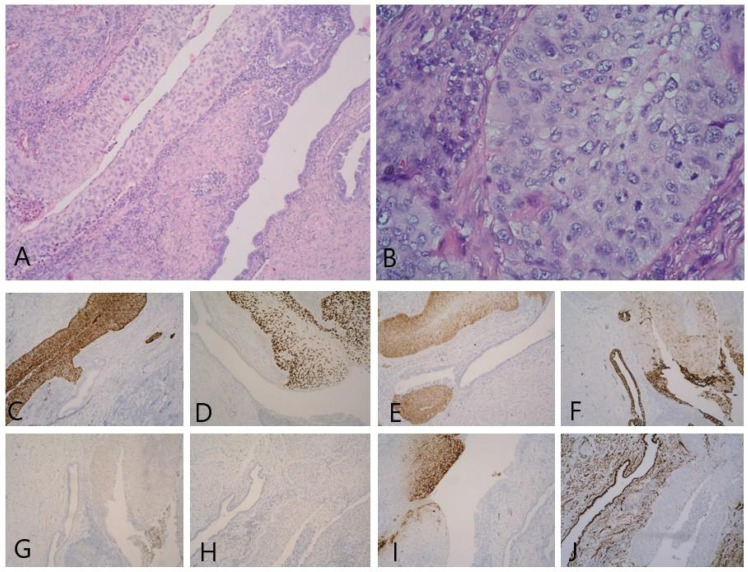
(**A**) Fallopian tube with colonization in part of the epithelium by tumor cells XE, 100×; (**B**) the tumor cell population showing distinct signs of cellular atypism XE, 100×; (**C**) CK5/6—positive in tumor cells, 100×; (**D**) p63—nuclear IHC expression in tumor cells, 100×; (**E**) p16—moderate-intensity nuclear positivity and weak cytoplasmic IHC expression in tumor cells, 100×; (**F**) CK7—positive in tumor cells, 100×; (**G**) CK20—weak IHC expression in tumor cells, 100×; (**H**) GATA 3—negative 100×; (**I**) CEA—positive in tumor cells, 100×; (**J**) WT-1—negative in tumor cells against a background of WT-1-positive stroma and tubular epithelium, 100×.

**Figure 5 reports-06-00054-f005:**
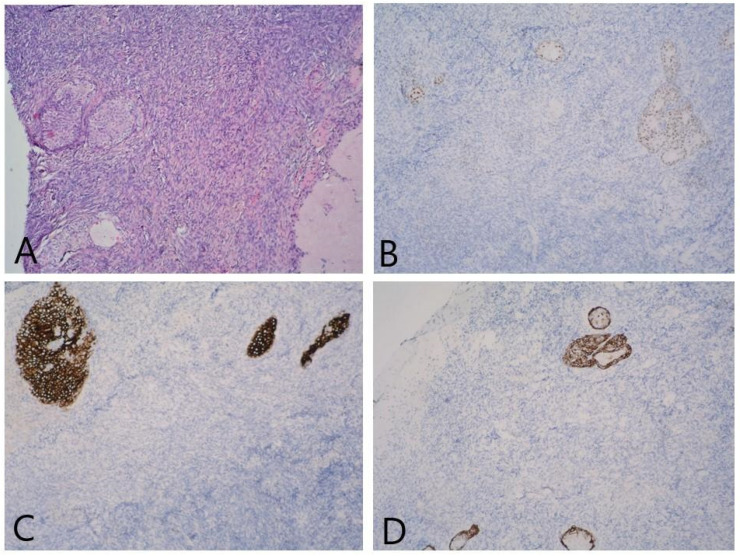
(**A**) Against the background of ovarian stroma and corpus albicans, there are soft nests of cells with a morphology similar to transitional epithelium (benign Brenner tumor) XE, 100×; (**B**) GATA 3—weak to moderate nuclear IHC expression in Brenner tumor cells, 100×; (**C**) CK7—positive in Brenner tumor cells, 100×; (**D**) CK5/6—positive in Brenner tumor cells, 100×.

**Table 1 reports-06-00054-t001:** The site of origin, spread/involvement, and immunoprofile of primary “transitional cell carcinomas”/“squamous-urothelial carcinoma”/“Papillary squamotransitional carcinoma” of the female reproductive system.

Author/Article	Type of Reported Tumor (as Reported)	Primary Site	Site(s) Involved	Positive Expression	Negative Expression
Lininger RA et al. [17]	transitional cell carcinomas	transitional cell carcinomas can affect the ovaries, fallopian tubes, cervix, and endometrium	tumors originating from the endometrium can metastasize to the ovaries, ovarian transitional cell carcinoma may metastasize to the myometrium	CK 7 (some cases might be negative)	CK 20
Patrelli TS et al. [18]	squamous urothelial carcinoma	vagina	one site was involved (no spread of the disease)	CK 7, p16 ink4a, p63	CK 20
Tong J et al. [19]	transitional cell carcinoma	endometrium (arising in a polypoid adenomyoma)	one site was involved (urinary tract carcinoma was excluded)	CK AE1/AE3, CAM5.2, CK7, EMA, PAX-8, CD56	CK20
Gitas et al. [8]	papillary squamotransitional carcinoma	uterine cervix	ovarian involvement	CK7, p63, CK5, CK14, p16	CK20 and ER
Lee DH et al. [20]	transitional cell carcinoma	fallopian tube	pelvic rectal peritoneum involvement	CK7 and EMA	p53, CK20, p63

## Data Availability

The authors declare that all related data are available upon request via the corresponding author’s email.

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
