# Peer review of "Squamotransitional Cell Carcinoma of the Uterine Cervix with Ovarian Metastasis and Benign Brenner Tumor: A Case Report"

_reports, 2023, doi:10.3390/reports6040054_

Round 1

Reviewer 1 Report

Comments and Suggestions for Authors

Potentially interesting case report. However, there are non-convincing elements which strongly decrease value of revised paper and lower rating of manuscript.

1. Substantially, what was the rationale of the blind abrasion of the uterus in case of AUB, D&C shouldn't be performed... Nowadays in such cases gold standard is diagnostic HSC. In such situation we would have more important data... Mostly about precise location of the lesions.

2. there are used unexplained acronyms.

3. what exactly was performed? lsc? meigs-wertheim?

4. What was the rationale for exactly such set of IHC markers? CK 5/6; 7 and 20 whereas the authors didn't perform e.g. Ki-67. steroid receptors. We should not use terms; "staining" instead of IHC expression.  

5. there are references which might be refreshed as well 7 and 19 are the same...

6. there is a misplaced and misleading localization of the ref. in the text, to be verified. As well there are 26 pos. while in ref. there is 25.

7. how do we know what was the direction of movement of tumor? Possibly we may have Brenner which transform into malignant form as transitional epithelium is very typical for this tumor... How do we know that it was not synchronic tumor of the ovaries with transformation of one of them? How do we know that it originates from cervix? Did the authors were thinking about molecular testing lesions?

Comments on the Quality of English Language

Minor editing of English language required. Acronyms should be dismantled, especially in the first use. 

Author Response

Potentially interesting case report. However, there are non-convincing elements which strongly decrease value of revised paper and lower rating of manuscript.

Authors reply: Thank you for your time to review this manuscript

  1. Substantially, what was the rationale of the blind abrasion of the uterus in case of AUB, D&C shouldn't be performed... Nowadays in such cases gold standard is diagnostic HSC. In such situation we would have more important data... Mostly about precise location of the lesions.

Authors reply: You are right but during the diagnosis the patient has severe bleeding so hysteroscopy was impossible to be done.

  1. there are used unexplained acronyms.

Authors reply: You are right, we fixed it

  1. what exactly was performed? lsc? meigs-wertheim?

Authors reply: The operative procedure is type C2 radical hysterotomy according to Querleu–Morrow Classification of Radical Hysterectomy - 2017

  1. What was the rationale for exactly such set of IHC markers? CK 5/6; 7 and 20 whereas the authors didn't perform e.g. Ki-67. steroid receptors. We should not use terms; "staining" instead of IHC expression.

Authors reply: These IHC markers are used because they are accepted for the diagnose of squamotransitional carcinoma.  We replace staining with IHC expression. 

  1. there are references which might be refreshed as well 7 and 19 are the same...

Authors reply: You are right.  We fixed it.

  1. there is a misplaced and misleading localization of the ref. in the text, to be verified. As well there are 26 pos. while in ref. there is 25.

Authors reply: You are right.  We fixed it. This is a simple mistake – reference 26 is 25

  1. how do we know what was the direction of movement of tumor? Possibly we may have Brenner which transform into malignant form as transitional epithelium is very typical for this tumor... How do we know that it was not synchronic tumor of the ovaries with transformation of one of them? How do we know that it originates from cervix? Did the authors were thinking about molecular testing lesions?

Authors reply: The cervical lesion was significantly larger, compared to the uterine extension and the lesions in the fallopian tube and ovary. Tumor emboli were evident in and around the cervical tumor (but not in the ovary). Brener tumors have distinct GATA3 positivity (demonstrated in the case report) and the caner component was clearly negative (in all individual foci of cancer). In adition as added in the text:” It is common knowledge in clinical practice that transitional cell carcinoma of the endometrium can coexist with ovarian Brenner's tumor, raising the possibility that there may be some relationship between the two. It is hypothesized to be either a result of “multicentric metaplastic process” (neometaplasia), affecting ovarian coelomic epithelium and Mullerian uterine epithelium, or consequence of the so called „field effect" – site determined presentation of an entity. “(ref. Giordano G, D'Adda T, Gnetti L, Merisio C, Raboni S. Transitional cell carcinoma of the endometrium associated with benign ovarian Brenner tumor: a case report with immunohistochemistry molecular analysis and a review of the literature. Int J Gynecol Pathol. 2007;26(3):298-304. doi: 10.1097/01.pgp.0000236943.28969.b2.) Genetic testing on one single case may give inconclusive and confusing results and was not done in this report (still it might be a good idea).

Reviewer 2 Report

Comments and Suggestions for Authors

Thank you for giving me a good opportunity to review this manuscript. I think it is interesting topic. It is better to correct some issues before publication

1. There are many abbreviations in manuscript. Please write full term and use abbreivations.

   For example, CC, PSCC, RMS, AH, WHO...etc.

2. Why the authors use TNM staging instead of the FIGO staging?

    It is not usual to use TNM staging in gynecologic cancer.

3. Please proofread your English grammar to make it more understandable.

Comments on the Quality of English Language

Please proofread your English grammar to make it more understandable.

Author Response

Thank you for giving me a good opportunity to review this manuscript. I think it is interesting topic. It is better to correct some issues before publication

 Authors reply: Thank you for reviewing this manuscript.

  1. There are many abbreviations in manuscript. Please write full term and use abbreivations.

   For example, CC, PSCC, RMS, AH, WHO...etc.

Authors reply: You are right. We fixed it.

  1. Why the authors use TNM staging instead of the FIGO staging?

    It is not usual to use TNM staging in gynecologic cancer.

Authors reply: In Bulgaria we used and both classification so we added and FIGO classification in the text.

  1. Please proofread your English grammar to make it more understandable.

Authors reply: Done as recommended.

Reviewer 3 Report

Comments and Suggestions for Authors

Squamotransitional cell carcinoma is a rare entity and will be interesting to report. My suggestions:

1. First two paragraphs of the "case presentation" part was too long and sometimes confusing? For example, what is RMS, is it relevant to the case? Please summarize imaging and clinical findings in a more organized way instead of listing them somewhat randomly.

2. I'm not sure what is "white bodies", do you mean "corpus albicans"?

3. Somewhere in the papar, the authors should explicitly explain what are the morphological and/or immunohistochemical features deem a SCC a squamotransitional cell carcinoma. 

4. For the discussion, instead of listing IHC profiles from each paper, please summarize in words or list in a table. The current form is too tedious and hard to read.

Comments on the Quality of English Language

The English language is in general fine. Moderate modification is needed to make the paper more precise.

Author Response

Squamotransitional cell carcinoma is a rare entity and will be interesting to report. My suggestions:

  1. First two paragraphs of the "case presentation" part was too long and sometimes confusing? For example, what is RMS, is it relevant to the case? Please summarize imaging and clinical findings in a more organized way instead of listing them somewhat randomly.

Authors reply: Done as recommende

  1. I'm not sure what is "white bodies", do you mean "corpus albicans"?

Authors reply: You are right. We fixed it.

  1. Somewhere in the papar, the authors should explicitly explain what are the morphological and/or immunohistochemical features deem a SCC a squamotransitional cell carcinoma.

Authors reply: Please, find the answer in the edited text:

Our data supports the statement by Patrelli TS et al. that leading for the diagnosis are the morphology and  cytoarchitectural features of this type of tumors. The  morphological findings are consistent with  mitotically active tumor with solid and/or inverted papillary growth pattern, composed of polygonal cells demonstrating transitional cell features (with hyperchromatic, oval, grooved nuclei and relatively small amount of cytoplasm) showing focal squamous differentiation. Additionally,  immunohistochemical staining with CK7 (usually positive) and CK20 (always negative) are good auxiliary markers for the diagnostic process. The authors also suggest other immunomarkers including p63 and  p16ink4a for the differential diagnosis.   (Patrelli TS, Silini EM, Berretta R, Thai E, Gizzo S, Bacchi Modena A, Nardelli GB. Squamotransitional cell carcinoma of the vagina: diagnosis and clinical management: a literature review starting from a rare case report. Pathol Oncol Res. 2011;17(1):149-53. doi: 10.1007/s12253-010-9280-8. Epub 2010 May 30. PMID: 20512667.) Among the reviewed articles, p63, Ck5, EMA and p16 were more often positive (but data is limited to just several studies)

( Patrelli TS, Silini EM, Berretta R, Thai E, Gizzo S, Bacchi Modena A, Nardelli GB. Squamotransitional cell carcinoma of the vagina: diagnosis and clinical management: a literature review starting from a rare case report. Pathol Oncol Res. 2011;17(1):149-53. doi: 10.1007/s12253-010-9280-8.; Tong J, Kasznica J, Habib F. Transitional cell carcinoma of endometrium arising in a polypoid adenomyoma in the endometrial cavity. Am J Clin Pathol. 2013; 140(suppl.1):A188, https://doi.org/10.1093/ajcp/140.suppl1.188.; Gitas G, Ertan K, Rody A, et al. Papillary squamotransitional cell carcinoma of the uterine cervix: a case report and review of the literature. J Med Case Reports 2019:13: 319. https://doi.org/10.1186/s13256-019-2217-2; Lee DH, Cho DH, Kim KM, Yim CY, Lee NR. Primary transitional cell carcinoma of the fallopian tube: A case report and literature review. Medicine (Baltimore). 2020;99(22):e20499. doi: 10.1097/MD.0000000000020499.)

Analyses based on larger populations demonstrate the almost persistent CK7 positive, CK20 negative (with rare exceptions) profile of squamotransitional cell carcinomas of the uterine cervix. (Koenig C, Turnicky RP, Kankam CF, Tavassoli, FA. Papillary squamotransitional cell carcinoma of the cervix: A report of 32 cases. The American Journal of Surgical Pathology 1997;21(8):915-921.)”

  1. For the discussion, instead of listing IHC profiles from each paper, please summarize in words or list in a table. The current form is too tedious and hard to read.

Authors reply: Done as recommende - we added table 1

Round 2

Reviewer 1 Report

Comments and Suggestions for Authors

I see clearly improvement of resubmitted manuscript.

I got acquainted with responses of the authors.

- I don't agree with reason of absence of Ki-67 and steroid receptors staining. It is deeply informative in reproductive system specimens, whereas proliferation index is routinely performed.

- Querleu–Morrow Classification - it should be placed in Ref. or shown in table. Eg. https://pubmed.ncbi.nlm.nih.gov/28785898/

- Explanation in point 7 is acceptable and clear.

- Minor editorial mistakes to be fixed easily -

line 102 "immunpro...."

usage of "x" letter instead of multiplication sign.

usage of PSCC acronym in line 123 without explanation.

"Immunostains" instead of expression in the table 1. 

Author Response

I see clearly improvement of resubmitted manuscript.

Authors reply: Thank you for your time to review this manuscript

I got acquainted with responses of the authors.

- I don't agree with reason of absence of Ki-67 and steroid receptors staining. It is deeply informative in reproductive system specimens, whereas proliferation index is routinely performed.

Authors reply:  We included two more pictures in figure 2 and added the following text:

  1. K) ER negative in the the tumor cells in the context of positive stromal cells, ER, 400x; L)Ki-67 - high proliferation rate in the invasive tumor nests Ki-67, 400x.

- Querleu–Morrow Classification - it should be placed in Ref. or shown in table. Eg. https://pubmed.ncbi.nlm.nih.gov/28785898/

Authors reply: Done as recommended.

- Explanation in point 7 is acceptable and clear.

Authors reply: Thank you

- Minor editorial mistakes to be fixed easily -

line 102 "immunpro...."

usage of "x" letter instead of multiplication sign.

usage of PSCC acronym in line 123 without explanation.

"Immunostains" instead of expression in the table 1.

Authors reply: All mistakes are fixed

Reviewer 3 Report

Comments and Suggestions for Authors

Page 2 line 49. Please add what morphological and immunohistochemical features define squamotransitional CC here. The authors seem to beat around the bushes with this definition. A precise and succinct definition will help the readers to understand the entity and the paper much better, and it is better to do this in the introduction. I understand that for each entity there are additional nuances to the classic picture, which can be discussed in details in the discussion.

Page 2 line 73. Please list the IHC profile of the tumor, that way the readers could refer to the definition of the entity to recognize and agree that this is a squamotransitional CC. The figure should not replace this list, as there might be more IHCs performed than displayed in the figure, and it is much harder to comprehend the figure than reading a list. Also, please consider add IHC names to each panel in the figure for easier reading.

Page 5 line 123, please define PSCC. The term appeared in conclusion as well without definition. Please be consistent with which term to use for the tumor. 

Comments on the Quality of English Language

No comment

Author Response

Page 2 line 49. Please add what morphological and immunohistochemical features define squamotransitional CC here. The authors seem to beat around the bushes with this definition. A precise and succinct definition will help the readers to understand the entity and the paper much better, and it is better to do this in the introduction. I understand that for each entity there are additional nuances to the classic picture, which can be discussed in details in the discussion.

Authors reply: We added the following text in the end of the discussion part:

Finally, according to the contemporary WHO classification of the tumors of the uterine cervix, the squamotransitional carcinoma is a subtype of squamous cell carcinoma associated with HPV. The variant is defined as having "transitional- like appearance " but no immunoprofile is recommended. It is currently considered as histologic type from the past [28]. Still, the morphologic presentation of the lesion and the differencial diagnosis with involvement from transitional carcinoma originating from the urinary tract makes this morphologic pattern important for the clinical and diagnostic practice. No definitive immunoprofile is recommended for squamotransitional carcinoma but the above mentioned CK7+/CK20- profile is significant to distinguish from transitional carcinoma infiltration/metastatic involvement(see Table 1). Follow up of cases of squamous cell carcinomas with morphologic squamotransitional pattern is important to assure that such tumors have similar prognosis to other HPV related squamous cell carcinoma with no specific morphologic features and to monitor if the optimal therapy is applied

Page 2 line 73. Please list the IHC profile of the tumor, that way the readers could refer to the definition of the entity to recognize and agree that this is a squamotransitional CC. The figure should not replace this list, as there might be more IHCs performed than displayed in the figure, and it is much harder to comprehend the figure than reading a list. Also, please consider add IHC names to each panel in the figure for easier reading.

Authors reply: With all our respect we cannot fully agree with this statement. In the first variant of the manuscript the list of the IHCs was given in the text and the recommendation of the reviewer was to present it in that way.

Page 5 line 123, please define PSCC. The term appeared in conclusion as well without definition. Please be consistent with which term to use for the tumor.

 Authors reply: We fixed it – this is squamotransitional CC

Round 3

Reviewer 3 Report

Comments and Suggestions for Authors

Recommend to accept